# Hydrogen Sulfide in Bone Tissue Regeneration and Repair: State of the Art and New Perspectives

**DOI:** 10.3390/ijms20205231

**Published:** 2019-10-22

**Authors:** Laura Gambari, Brunella Grigolo, Francesco Grassi

**Affiliations:** Laboratorio RAMSES: IRCCS Istituto Ortopedico Rizzoli, Via di Barbiano 1/10, 40136 Bologna, Italy; laura.gambari@ior.it (L.G.); brunella.grigolo@ior.it (B.G.)

**Keywords:** hydrogen sulfide, bone tissue regeneration, osteogenesis, vasculogenesis, inflammation, cell recruitment, endochondral ossification, H_2_S-releasing scaffolds, tissue engineering

## Abstract

The importance of hydrogen sulfide (H_2_S) in the regulation of multiple physiological functions has been clearly recognized in the over 20 years since it was first identified as a novel gasotransmitter. In bone tissue H_2_S exerts a cytoprotective effect and promotes bone formation. Just recently, the scientific community has begun to appreciate its role as a therapeutic agent in bone pathologies. Pharmacological administration of H_2_S achieved encouraging results in preclinical studies in the treatment of systemic bone diseases, such as osteoporosis; however, a local delivery of H_2_S at sites of bone damage may provide additional opportunities of treatment. Here, we highlight how H_2_S stimulates multiple signaling pathways involved in various stages of the processes of bone repair. Moreover, we discuss how material science and chemistry have recently developed biomaterials and H_2_S-donors with improved features, laying the ground for the development of H_2_S-releasing devices for bone regenerative medicine. This review is intended to give a state-of-the-art description of the pro-regenerative properties of H_2_S, with a focus on bone tissue, and to discuss the potential of H_2_S-releasing scaffolds as a support for bone repair.

## 1. Introduction

Tissue development and repair has long been associated with sulfur metabolism: The first evidence dates back to the study by Williamson and Fromm (1955) which found an association between the activation of sulfur-containing amino acids metabolism and the wound healing process [1]. However, more than six decades have passed before the first studies linking biological functions of sulfurated amino acids to hydrogen sulfide (H_2_S) came to light. Notably, the identification (1996) of a physiological role of H_2_S in the brain tissue [2] opened the way to several studies unravelling its multiple biological functions. H_2_S is a gaseous molecule produced endogenously by the enzymes cystathionine-β-synthase (CBS), cystathionine-γ-lyase (CSE) and, to a lower extent, 3-mercaptopyruvate sulfurtransferase (MPST) [3], within the transsulfuration pathway. A broad range of studies further clarified that H_2_S is not only a “secondary reaction product” of this pathway but is a critical mediator of physiological functions in many tissues.

Over the past years, several therapeutic applications of sulfur-containing compounds or H_2_S-releasing donors have emerged and a few clinical trials are ongoing [4]. However, while the studies on the systemic use of H_2_S-donors are at an advanced stage, the need for obtaining an organ-specific delivery of H_2_S to exploit the tissue-specific cytoprotective and regenerative properties of H_2_S has only recently emerged [5]. One field where H_2_S-releasing biomaterials hold a great potential interest is bone repair and regeneration.

Bone is a vascularized, dynamic tissue which provides the body with structural and movement support functions, organ protection, mineral reserve, and endocrine system function [6,7]. It is continuously remodeled by the coordination and balance of two key processes, bone formation and bone resorption [8,9], and is capable of repairing and reshaping without leaving scars [10]. However, bone lesions caused by trauma and pathological bone resorption (infectious diseases, biochemical disorders, congenital disorders, or abnormal skeletal development) have limited capacity to heal and require large amounts of autologous/allogenic bone or bone substitutes for reconstruction [11,12]. In recent years, research approaches have moved from a focus on inert biomaterials toward the complex design of biomaterials mimicking the structure of the native bone extracellular matrix and being able to provide a controlled release of biomimetic, pro-osteogenic factors [13].

Over the past few years, it has become increasingly clear that H_2_S regulates bone homeostasis and that systemic administration of H_2_S holds good therapeutic potential, as it prevented or reversed pathological bone loss [14,15,16]. However, whether H_2_S affects the processes of bone tissue regeneration and repair and whether it could be used as a biomimetic factor in regenerative medicine approaches has not been fully elucidated. This review is intended to first summarize the emerging evidence linking H_2_S and bone tissue repair and regeneration. Then, the state-of-the-art of the development of H_2_S-releasing scaffolds and biomaterials for promoting tissue regeneration will be described, with a focus on bone repair. Finally, implications and future challenges in the field of H_2_S-releasing scaffold for bone tissue engineering (BTE) will be discussed.

## 2. Bone Tissue Repair: H_2_S as a Suitable Biological Cue for Scaffold Functionalization

Bone remodeling is an orchestrated process constantly occurring to let the skeleton adapt to biomechanical loading, microfractures, environmental stress [17], and to control calcium and other ions levels in the circulation. Similarly, bone repair occurs in response to injury by activating complex and orchestrated regenerative processes to restore bone tissue integrity, structure, and homeostasis [18,19]. Despite the high innate regenerative capacity of bone, critical-size defects and non-unions fail to heal, thus making them a global health burden and a major challenge in regenerative medicine [12,20,21,22]. Autograft is currently the gold standard therapy and achieve an 80% success rate [23]. However, the percentage of non-union remains significantly high and the procedure involves two surgical operations and donor site morbidity for patients, thus conditioning their quality of life. A wide variety of bone substitutes have been tested in clinics to improve bone repair, among which are allogenic bone and natural or synthetic osteoconductive matrices [24,25], obtaining promising results. In this context, BTE offers the opportunity to develop innovative alternative treatment options that will ideally eliminate the limitations of current treatments. BTE is based on various combinations of three principal components: biomaterials as scaffolds, regulatory signals such as growth factors (GF), and cells. Recently, the design of functionalized cell-free scaffolds has emerged as a useful tool to overcome current drawbacks linked to cell-based approaches (limited autologous cells, time/cost-intensive cell expansion procedures, relative low cell survival rate, and high risk of immune-rejection) [26]. Strategies of scaffold functionalization involve GF, chemokines, and peptides that are able to activate bone resident cells and promote bone regeneration [26]. These biological cues are usually incorporated into scaffolds via physical adsorption, chemical covalent coupling, and encapsulation [26], and their controlled and site specific release is crucial for the success of tissue engineering strategies. Over the past 50 years, several studies aimed at understanding the biological processes which takes place in in vivo models of bone fractures during bone repair [27], particularly in in the presence of bone fractures [28,29], and allowed the identification of several biological cues. Accordingly, bioinspired and biomimetic cell-free scaffolds were recently developed to mimic the natural self-healing events of bone healing by using biological signals able to modulate inflammation and induce cell recruitment, vascularization, and osteogenic differentiation [26]. Figure 1 summarizes the biological processes involved in bone fracture repair, highlighting which of them may be regulated by H_2_S (discussed more in detail in the next sections).

### 2.1. H_2_S Regulates Inflammation in Bone Tissue

Recent investigations in the field of osteoimmunology have demonstrated extensive cross-talk between immune and skeletal systems [30]. In particular, upon bone injury, hematoma formation is closely followed by acute inflammation and release of inflammatory cytokines which activate a cascade of processes to promote bone healing. A tightly regulated balance between pro-inflammatory macrophages (M1) and pro-healing, anti-inflammatory macrophages (M2) results in inflammatory and mesenchymal stromal cells (MSCs) recruitment, promotion of neovascularization, and induction of MSCs towards osteogenesis [30]. Notably, their absence has been shown to completely abolish callus formation [31]. Treg also promotes tissue healing by inducing M2 polarization, inhibiting neutrophil infiltration, activity, and inducing their apoptosis [32], supporting MSCs differentiation [33] and inhibiting osteoclasts (OCs) differentiation [32]. The ablation of δγ-T-cells promotes bone healing since mice deficient in this cellular type have a shorter time to fracture union and show improved biomechanical strength compared to control mice [34]. Conversely high presence of T-cells promotes OCs differentiation and inhibit MSCs [35] causing additional “by-stander” tissue damage [30]. Therefore, scaffold functionalization strategies are currently aimed at modulating pro-inflammatory and anti-inflammatory activity, thus preventing chronic inflammation and unbalanced inflammatory responses responsible for the formation of fibrous capsules around scaffolds, which make scaffolds bioinert [36].

H_2_S plays an important role during inflammation in several tissues [37] and the increased H_2_S levels occurring during inflammation have been proposed as a stress-response mechanism to the injury caused by oxidative stress [38]. In OCs, inhibition of reactive oxygen species (ROS) signaling by H_2_S occurs through direct and indirect mechanisms by triggering NRF-2-dependent antioxidant response and thus inhibiting osteoclast differentiation (Figure 2) [39]. Interestingly, H_2_S was shown to be a key mediator in the immediate anti-inflammatory and antioxidant function of N-acetyl-cysteine, a broadly used mucolytic medicine [40]. H_2_S was shown to reduce edema formation [41], promote neutrophil apoptosis, suppress the expression of some leukocyte and endothelial adhesion molecules [42], stimulate macrophages polarization to M2 [43,44,45], and activate the AnxA1 pro-resolutive pathway [46]. Concerning bone tissue, in the animal model of hyperhomocysteinemia (CBS^−/−^ mice), the autosomal recessive disease involving CBS, higher levels of inflammation has been correlated to lower bone tissue [16]. Particularly, these mice had increased acetylation of NF-kB p65, resulting in increased levels of IL-6 and TNF-α in the circulation and inhibition of Runx-2 and Ocn gene expression [16]. Administration of NaHS suppressed histone acetylation dependent NF-kB p65 signaling activation and inflammation and rescued osteogenic genes expression [16]. Moreover, H_2_S exogenous administration (by using the H_2_S-releasing moiety of ATB-346 and ATB-352 molecules) has been used to inhibit inflammation and inflammatory bone loss in experimental periodontitis [47,48]. A therapeutic platform has been developed to deliver H_2_S and inhibit P-selectin expression in human platelets, which play an essential role in the initial recruitment of leukocytes to the site of injury during inflammation [49]. Moreover, H_2_S has been shown to mediate the inhibition of inflammatory responses during bacterial infections in an ex vivo model of cells infected with mycoplasma [50]. The molecular pathways imply inhibition and activation of nuclear translocation of NF-κB, reducing the transcription of pro-inflammatory genes [50]. Finally, H_2_S plays a role in mediating both innate and adaptive immunity, since it was also found to be essential for Treg cell differentiation and function and enhances T-cell proliferation, activation, and cell death [51,52]. Moreover, CBS^−/−^ mice developed autoimmune disease [53].

In tissues other than bone, H_2_S has been shown to improve wound healing by attenuating inflammation and increasing angiogenesis. H_2_S has been hypothesized to play a role in severe burns, which are associated with processes that causes increased permeability and edema [54], and has been proven to improve diabetic wound healing [55,56].

Taken together these data indicates that future design of H_2_S-releasing scaffolds for BTE may be tuned to temporally modulate pro-inflammatory and anti-inflammatory activities, thus promoting the formation of regenerated bone tissue.

### 2.2. H_2_S Regulates Migration and Survival of Cells Involved in Bone Repair

The recruitment of autogenous endothelial progenitor cells (EPCs) and MSCs to the site of bone defect is critical to induce bone tissue repair [57]. More in general, EPCs play a crucial role in endothelial restoration after arterial injury [58], and the migratory potential of MSCs has been demonstrated to be one of the most important factors for neovascularization and regeneration [59,60]. H_2_S has been found to induce migratory properties in both cell types. In particular, NaHS, an H_2_S donor, increases the mobilization of EPCs after vascular injury, enhancing their adhesion and colony formation capacities [61] and improving re-endothelization in nude mice with carotid artery injury [62]. CSE-mediated H_2_S deficiency in high glucose-treated MSCs impairs their migratory capacity, which was further restored by NaHS treatments [63]. Furthermore, H_2_S has been found to mediate the migratory potential of other cells types. In particular, NaHS accelerates skin fibroblasts and human keratinocytes in migration assays [64], thus enhancing wound healing. H_2_S promotes macrophage migration to an infarcted area, by accelerating internalization of integrin β1 and activating the downstream Src-FAK/Pyk2-Rac pathway, thus promoting infarction healing [65].

Moreover, it is well known that H_2_S promotes proliferation and survival of MSCs and of other cell types during several stresses such as hypoxia, oxidative damage, and serum deprivation. Particularly, it inhibits cellular death by apoptosis [66,67]. Therefore, H_2_S may act similarly to other GF, which initiate the repair process by facilitating proliferation and differentiation of the stem cells that give rise to the fracture callus [68].

Whether H_2_S modulates cells migration, survival, and proliferation to the bone defect to induce bone tissue repair is still an open question.

### 2.3. H_2_S Promotes the Expression of Angiogenic Factors in Bone Tissue

Vascularization is a primary driving force behind both intramembranous and endochondral ossification. To achieve a highly functional new regenerated tissue, the tight connection of new vasculature formation and osteogenesis is of utmost importance. New vessels are fundamental both in the initial and in the final step of bone repair. In the first phase, new blood vessels are essential to provide nutrients and facilitate cells migration toward the damaged site and in the final phases of repair, vascular remodeling occurs and vessels regresses to the original state [69]. Importantly, failure of angiogenesis can lead to nonunion [70]. Deletion of vascular endothelial growth factor (VEGF) in osteoblasts (OBs) interrupts the coupling of osteogenesis and angiogenesis, and delays the intramembranous ossification-mediated repair during cortical bone defect healing [71]. To date, there is no information linking H_2_S promotion of neo-angiogenesis during bone regeneration and repair. However, several evidences may suggest a role. H_2_S has been shown to stimulate in vitro the three processes crucial for the neo-angiogenesis: proliferation, migration, and tube-like network formation by endothelial cells [72]. Evidence of neo-vascularization of Matrigel implants in vivo further confirmed the ability of H_2_S in promoting new blood vessel formation [73,74]. These features are at least partly mediated by H_2_S-dependent activation of two factors critical for bone vascularization: hypoxia-inducible factor-1 (HIF-1α) and VEGF. Indeed, a number of studies have established extensive crosstalk between H_2_S and VEGF. Exogenously administered H_2_S has been shown to upregulate VEGF expression in several tissues and cells [75], among which skeletal muscle cells and bone cells in vitro and in vivo [76,77]. H_2_S also significantly upregulated HIF-1alpha protein levels and increased HIF-1alpha binding activity under hypoxic conditions [78]. Finally, H_2_S improved vascular density and blood flow in ischemic skeletal muscles of CBS^+/−^ mice or diabetic type-2FAL mice models with hindlimb femoral artery ligation [63,79].

### 2.4. H_2_S Stimulates Bone Formation

Over the past few years, it has become increasingly clear that H_2_S affects bone tissue regeneration by acting at several levels such as regulation of bone cells activity, reduction of oxidative stress, regulation of calcium intake by bone cells, and promotion of angiogenesis. CBS and CSE, the H_2_S-producing enzymes, are expressed both in MSCs [15,80] and in OBs [80,81,82]. In particular, CSE is the predominant source of H_2_S in OBs [81]. H_2_S plays a cytoprotective role in bone cells; it protects OBs against homocysteine-induced mitochondrial toxicity [83]; and OBs against hydrogen peroxide (H_2_O_2_)-induced cell death and apoptosis [84].

H_2_S was shown to play an important role in sustaining osteogenic differentiation of human MSCs (hMSCs) [14,15] and in modulating OCs differentiation [39,85,86,87]. CBS and CSE levels increased during osteogenic differentiation and play an essential role in osteogenic differentiation in vitro [15,80]. Notably, H_2_S-deficiency due to CBS knock-down was linked to impaired osteogenesis and bone loss in mice [15]. Several pathways are implicated in H_2_S-dependent induction of osteogenesis, summarized in Figure 2: the WNT pathway [14,88]; BMP pathway [77]; and p38-MAPK and ERK signaling pathways [84,89]. Moreover, S-sulfhydration of cysteine residues in critical proteins involved in osteogenesis, such as Runx-2 [81] and Ca^2+^ TRP channels [15], has been shown to be a mechanism of H_2_S-dependent induction of osteogenesis. Furthermore, overexpression of CSE in OBs increased Runx-2 *S*-sulfhydration and enhanced OBs biologic function (increased ALP activity, Alizarin red-positive calcification nodules, and osteogenic gene expression). Interestingly, mechanical loading in periodontal tissues up-regulated CSE expression and OCs formation inducing bone remodeling and accelerating orthodontic tooth movement (OTM) speed. Similarly, OTM and bone remodeling are stimulated by the up-regulation of osteoclastogenesis and osteoblastogenesis induced by H_2_S exogenous administration [90].

An imbalance in H_2_S metabolism is associated with defective bone homeostasis. Evidence from several preclinical models showed that the depletion of H_2_S levels is implicated in bone loss; data were reported in ovariectomized mice [14,91], in H_2_S-deficient CBS^+/−^ mice [15], in a model of H_2_S-deficiency linked to diet-induced hyperhomocysteinemia [16,92], and in methionine-restricted fed rats [93]. Interestingly, when the treatment with H_2_S donors in these models was investigated, finalized to normalize the plasma level of H_2_S, it was found that H_2_S was able to prevent or even reverse the bone loss. Moreover, asides from these conditions, H_2_S exogenous administration has been shown to have therapeutic potential against bone loss induced by modelled microgravity [94] and other conditions such as distraction osteogenesis [95,96]. Overall, these data demonstrate that H_2_S regulates osteogenesis and bone formation in both healthy and pathological conditions.

In the context of bone repair, new bone formation occurs by direct and indirect mechanisms, involving intramembranous and endochondral ossification, as extensively reviewed elsewhere [97,98,99,100,101]. During these processes MSCs proliferate and differentiate into the osteogenic or chondrogenic lineages and increase the production of blood vessels from pre-existing vessels.

In models of fracture healing, treatment with H_2_S by various means affected both direct and indirect mechanisms of bone healing. In a mouse model of distraction osteogenesis, H_2_S injection was observed to accelerate the formation of mature microstructures of trabeculae and mineralization substituting the fibrous-connective tissue and improve the mechanical properties of the callus healing within the distraction gaps [95]. Moreover, in a rat model of intramedullary fixed fractural bone, Zheng et al. investigated the delivery of CSE adenovirus through absorbable gelatin. CSE overexpression resulted in improved bone fracture healing: less inflammatory infiltration, greater collagen expression, fibrocartilage, and osteocyte deposition. Interestingly, the authors showed increased endochondral ossification at the tissue repair site, and a full bridge between fracture sites in rat femurs [81].

A summary of the main studies investigating the effects of H_2_S-releasing donors on bone cells are reported in Table 1.

## 3. H_2_S-Releasing Scaffolds for Regenerative Medicine

Besides the efficacy of systemic administration of H_2_S, several reasons provide a rationale for the development of H_2_S-releasing scaffolds for bone regeneration.

First, cell-free scaffolds have undergone intense development in the recent years due to the drawbacks linked to the employment of cells in regenerative medicine [26]. Many physical, chemical, or biological functionalizations have led to tremendous progresses in the ability to support the physiological healing process of the bone [26]. For example, it was demonstrated that topographically defined implants are able to recruit osteoprogenitor cells at the site of bone injury and promote a more efficient repair in vivo [102,103]. Moreover, chemical modifications of the surface can endow biomaterials with increased osteoinductive capacity; for example, doping dense biomaterials with inorganic ions such as magnesium (Mg), Cobalt (Co), silicon (Sr), and Strontium (Sr) was shown to promote osteogenic differentiation in vitro and in vivo [104,105,106,107].

Second, the field of H_2_S-releasing materials is readily expanding. Several reports over the past few years showed promising results from in vitro and in vivo models and demonstrated that the biological properties of H_2_S can be successfully recapitulated in H_2_S-releasing devices for biomedical applications. In a recent report, Wu et al. successfully doped phosporodithioate donors (JK-1), a kind of H_2_S donors triggered by hydrolysis and releasing H_2_S in a pH-dependent fashion, with polycaprolactone (PCL) and the resulting fibers showing a slower kinetic of H_2_S-release compared to the native molecules in solution. When applied in an animal model of full-thickness skin damage, the doped PCL-JK1 fiber showed overall improved wound healing times as compared to non-doped fibers [108], thus confirming that H_2_S released by the fibers retained the previously evidenced properties of promoting wound healing [64,109,110]. The same authors developed a biomimetic hyaluronic acid (HY) hydrogel doped with JK1, to overcome the low permeability to oxygen of PCL fibers. This scaffold accelerated the wound repair process by decreasing inflammation, driving macrophages polarization toward M2, and increasing angiogenesis, and the effect was higher compared to adding JK1 to the wound microenvironment without being embedded in HY [45].

Nano-structured patches obtained by electrospun fibers are a valuable tool to promote tissue healing and to achieve local release of bioactive compounds [111]. Cacciotti et al. recently developed H_2_S-releasing poly(lactic) acid fibrous membranes functionalized with H_2_S donors of natural origin (from garlic oil-soluble extracts and diallyl disulfide) [112]. These donors where entrapped within the fibers and the H_2_S release by the membranes was prolonged and gradual and induces MSCs proliferation and anti-microbial activity. Following a similar approach, Feng et al. added the H_2_S donors (NSHd1) to PCL fiber spun by electrospinning to produce controlled, thiol-triggered H_2_S release [113]. Interestingly, by tuning the diameter of the fibers the authors achieved a controlled rate of H_2_S release. The authors showed that H_2_S increased gene expression of collagen type I and type III in fibroblasts while protecting them from the oxidative stress produced by H_2_O_2_, suggesting that these microfibers may support wound dressing. The group of Mauretti et al. developed a photopolymerizable PEG-fibrinogen hydrogel incorporating albumin microbubbles functionalized with thiosulfate cyanide sulfurtransferase, one of the enzymes able to catalyze the H_2_S-production [114]. H_2_S produced by the scaffold increased cardiac progenitor cell proliferation, thus retaining the feature previously described by others [115]. The group of Liang et al. loaded a partially oxidized alginate (ALG-CHO) with the H_2_S donor 2-aminopyridine-5-thiocarboxamide and tetraaniline (a conductive oligomer), and adipose-derived stem cells (ADSCs) [116]. The hydrogel, which mimics the slow and continuous release of endogenous H_2_S, increased the ejection fraction value and reduced the infarction size in rats with myocardial infarction, thus evidencing that H_2_S released retained the H_2_S-dependent protective role on myocardial infarction and heart failure previously described [115] and offering a promising therapeutic strategy for treating infarction.

Overall, the strategies to add H_2_S to the biomaterials described above were based on physical adsorption, chemical covalent coupling, and encapsulation. Beside these strategies, chemical research is also growing to develop methods for obtaining H_2_S-releasing donors with improved physical and chemical features. Of interest, an H_2_S-releasing gel was developed by the group of JB Matson, by using an H_2_S-releasing peptide that self-assembles into a robust hydrogel in water, obtained by chemical synthesis of an aromatic peptide amphiphile and the H_2_S moiety, S-aroylthiooxime—SATO—functional group. This H_2_S-releasing gel, showed a slow kinetic of H_2_S release and absence of cytotoxicity [117]. In a recent report, this H_2_S-releasing hydrogel substantially reduced intimal hyperplasia in human veins, showing efficacy at a dose five times lower than NaHS, a fast release H_2_S donor [118].

Altogether, these data evidence how H_2_S can be used to functionalize biomaterials inducing cell differentiation and function, and thus can be considered as a regulatory signal in the “tissue engineering triad” (Figure 3).

The only reported H_2_S-releasing biomaterial targeting bone regeneration was developed by our group [77]. Based on previous preclinical studies showing efficient stimulation of osteogenesis by H_2_S donors, we postulated that a local release of physiologically relevant H_2_S concentrations may trigger osteogenic differentiation of h-MSCs. An H_2_S-releasing silk fibroin (SF) sponge was developed by first applying a salt leaching approach to add appropriate porosity to the SF fibers and by then doping the H_2_S donor (GYY4137) to the sponges. The resulting SF-GYY scaffolds showed unaltered mechanical properties and no cytotoxicity compared to the native SF scaffolds. When tested in a model of 3D culture of h-MSCs within a perfusion bioreactor, H_2_S induced a significant increase in the differentiation to mature OBs, revealed by increased deposition of the mineral matrix (as shown in Figure 4) and increased expression of osteogenic genes after three weeks in culture. This study first provided a proof-of-principle that exposing h-MSCs to H_2_S released by a scaffold induced their differentiation toward OBs, and demonstrated that loading osteoconductive biomaterials with an H_2_S donor is a suitable strategy to promote OBs differentiation at sites of bone regeneration.

Overall, the evidences reported in the present review set H_2_S as an important emerging candidate for promoting regenerative medicine and particularly for BTE.

## 4. Perspectives for H_2_S-Loaded Scaffolds for BTE

Although research on the biological activities of H_2_S was first triggered by investigations in the field of neurology and circulation, a substantial body of evidence, summarized in this review, has recently shown that H_2_S also plays an important role in bone physiology and can stimulate the process of bone repair at multiple levels. The field of H_2_S-releasing scaffolds for BTE holds the promise to open new opportunities to treat bone tissue damages. However, it is still on its infancy and a number of challenges are still ahead of us, which will require joint efforts from material scientists, chemists, and biologists.

First, the range of biomaterials used for the combination with H_2_S is limited. Interestingly, all the H_2_S-releasing scaffold before mentioned (PCL, polylactic, PEG, alginate, and peptide amphiphilic) are made by biomaterials previously tested for bone regeneration alone or composite [119,120,121,122,123], making them of interest also in the field of BTE. However, further combination with other biomaterials, especially those which better mimics the composition and the biomechanical properties of bone, is necessary to take a step forward in this field.

Second, the design of new formulations of H_2_S donors should address the need for improved stability (e.g., in aqueous solution during scaffold development) and a “controlled” release of H_2_S (for avoiding the burst–release phenomenon). Notably, several formulations of H_2_S donors has been developed to release H_2_S triggered by a change in pH [124,125] or amounts of thiols, reactive oxygen species, or enzymes [5,126,127,128,129] within the microenvironment. These formulations partially addressed this need and may provide further opportunities to tune the H_2_S release in bone microenvironments according to different phases of bone healing to better exploit its biological properties. However, changes in the technologies for scaffolds development require a steady advancement in the development of H_2_S-donors. As an example, H_2_S donors are able to endure the high temperature required for gel printing while preventing premature H_2_S release would be of great interest given the rapid expansion of 3D printing technology for the development of critical-size bone defect customized products.

Third, new formulations of H_2_S donors should be tailored for applications in BTE. As an example, H_2_S donors able to bind with high affinity bone or bone substitutes, such as hydroxyapatite, would be useful for optimizing the interaction of the H_2_S donor within the scaffold matrices. The use of H_2_S-donors hybridized with bisphosphonates, molecules able to bind avidly bone, such as the recently developed H_2_S-releasing DM-22 [130], could represent an interesting approach to this challenge.

The knowledge generated by studies addressing these points will be instrumental for the development of H_2_S-releasing scaffolds tailored for bone regeneration and would greatly impact the advancement in the field of BTE. Furthermore, the direct correlation between bone healing and local H_2_S release by bone scaffolds will be fundamental to corroborate the rationale of H_2_S’s role in mediating bone repair.

## 5. Concluding Remarks

This review provides a concise and up-to-date overview of the role of H_2_S in bone tissue repair and of the ongoing research and future challenges for the development of H_2_S-releasing scaffolds for regenerative purposes. The research of new formulations of H_2_S-releasing scaffolds for BTE may provide patients with skeletal injuries with novel opportunities of treatment.

## Figures and Tables

**Figure 1 ijms-20-05231-f001:**
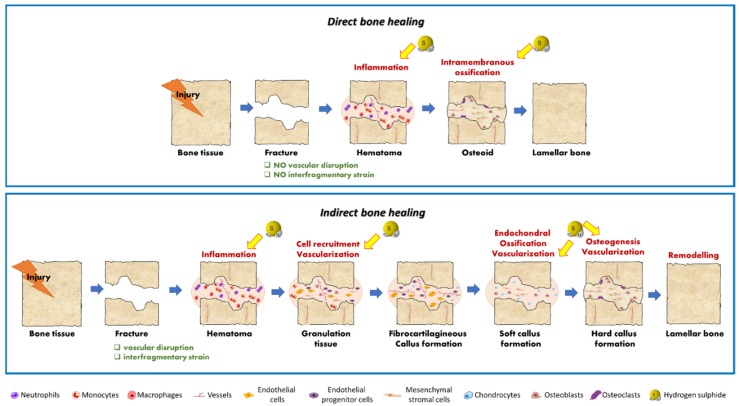
Schematic representation of the process of bone fracture healing. Immediately after bone injury, the healing process is initiated by hematoma formation and inflammatory response accompanied by the release of pro-inflammatory cytokines. In direct bone healing (occurring at fractures that are perfectly aligned and with no vascular disruption) the inflammation activates a process of intramembranous ossification which directly produces a new lamellar bone structure without requiring any remodeling step. In indirect bone healing (occurring in fractures with large, hypoxic defects and interfragmentary strains), the intramembranous ossification response occurs subperiosteally directly adjacent to the ends of the fracture, generating a hard callus. Adjacent to the fracture line, endochondral ossification occurs: A cartilaginous callus form and several cells migrates and proliferate within it, building an intermediate cartilaginous template which subsequently undergoes calcification (primary bone formation); the cartilage and primary mineralized matrix are then resorbed and secondary bone formation occurs; ultimately bone remodeling by osteoclasts and osteoblasts take place and the callus is totally replaced by new bone indistinguishable from adjacent, uninjured tissue. The processes by which H_2_S may modulate during bone fracture healing are highlighted by a yellow arrow.

**Figure 2 ijms-20-05231-f002:**
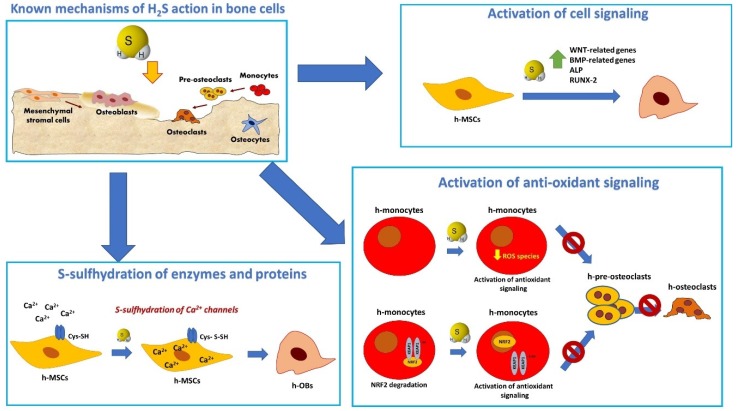
Schematic representation of the main mechanisms of action of H_2_S in mesenchymal stromal cells, osteoblasts, and osteoclasts. H_2_S modulates osteogenic differentiation in osteoprogenitor cells by inducing S-sulfhydration in TRP channels leading to increased Ca^2+^ influx in MSC [15]; at the transcriptional level, H_2_S donors induce the expression of osteogenic signaling through stimulation of multiple signaling pathway [14,77,88]. In the context of inflammation, H_2_S triggers an antioxidant signaling in OC which involves the increased nuclear translocations of NRF-2 and results in the inhibition of OC differentiation [39,87].

**Figure 3 ijms-20-05231-f003:**
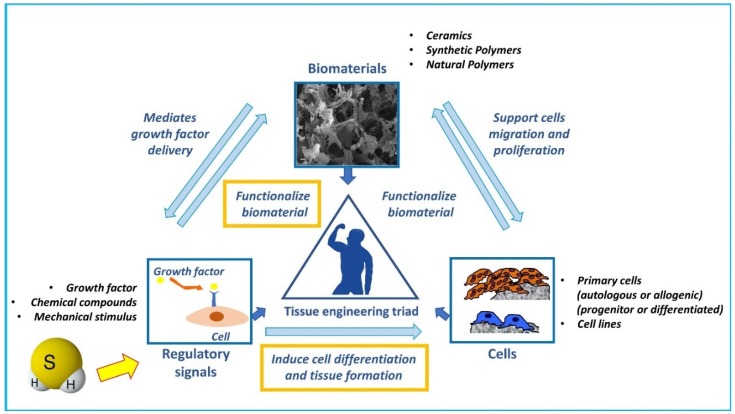
Schematic representation of potential H_2_S role in the “tissue engineering triad”. Three main factors are the key players of tissue engineering strategies and compose the so-called “tissue engineering triad”: 1) a scaffold that provides structure and support cells migration, colonization, proliferation, and differentiation, and is the substrate of loading of regulatory signals such as growth factors; 2) a source of cells to promote the required tissue formation (endogenous cells, primary cells, or cell lines pre-cultured or loaded within the scaffold); and 3) regulatory signals (growth factors, biophysical stimuli, etc.) to induce cell differentiation and tissue formation. We postulate that H_2_S may be considered as a regulatory signal able to functionalize biomaterials and induce endogenous or loaded cells differentiation and promote bone tissue formation.

**Figure 4 ijms-20-05231-f004:**
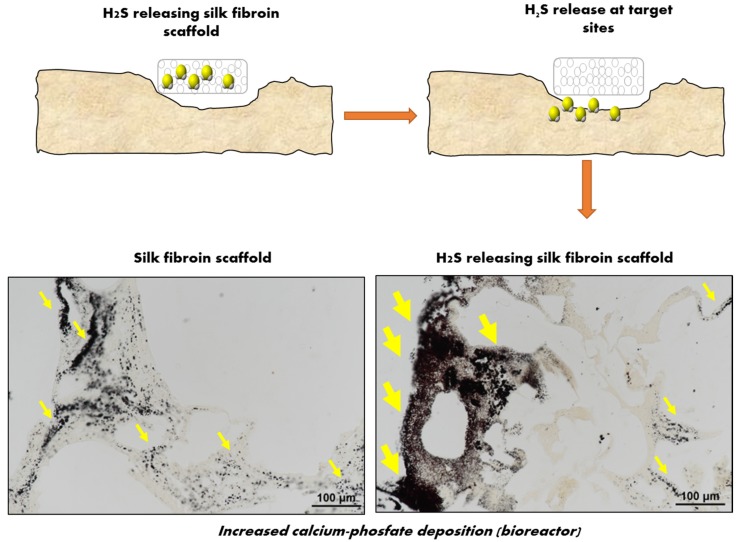
Schematic representation of the rationale for developing H_2_S releasing scaffolds. A silk fibroin (SF) porous scaffold loaded with the H_2_S donor GYY4137 (GYY) was developed with the rationale of providing the anabolic function of H_2_S to the damaged bone tissue. The combination of SF with H_2_S potentiated the osteoconductive properties of SF when h-MSCs where cultured in osteogenic media on a perfusion bioreactor. Cells remained viable and colonized the scaffolds. H_2_S activated genes and proteins involved in ossification, OBs differentiation, bone mineral metabolism, and angiogenesis, allowing a high and early mineralization (as shown by Von Kossa staining in the figure, yellow arrows indicate areas with more intense staining for mineralized matrix).

**Table 1 ijms-20-05231-t001:** Summary of the main studies and key findings on the regulation of bone cell function and bone metabolism by H_2_S-releasing agents.

Ref.	Authors	H_2_S-Donor	In Vitro/In Vivo	Species	Main Findings
[14]	Grassi et al.	GYY4137 NaHS	In vivo In vitro	Mice	GYY4137 prevented and reversed bone loss in ovariectomize mice. NaHS increased mRNA expression of BSP during osteogenic differentiation of hMSCs.
[15]	Liu et al.	GYY4137 NaHS	In vivo	Mice	NaHS improved BMMSC function from CBS^+/−^ mice both ex vivo and in vivo GYY4137 rescued osteopenia phenotype and BMMSC function in CBS^+/−^ mice
[16]	Behera et al.	NaHS	In vivo	Mice	NaHS ameliorates HHcy induced NF-κB acetylation dependent proinflammatory response in CBS^+/−^ mice NaHS inhibited osteoclast formation by monocytes derived from CBS^+/−^ mice NaHS induced osteogenic differentiation in BMMSC from CBS^+/−^ mice NaHS prevented bone loss in CBS^+/−^ mice
[75]	Gambari et al.	GYY4137 loaded in a SF scaffold	In vitro	Human	GYY4137 released by the SF scaffold induced the osteogenic differentiation of h-MSCs and the expression of integrins and angiogenic genes.
[81]	Zhai et al.	NaHS	In vitro	Murine cell line	NaHS protected by Hcy-mediated osteoblast mitochondrial toxicity and apoptosis.
[82]	Lv et al.	GYY4137	In vitro	Murine cell line	GYY4137 stimulated osteoblastic cell proliferation and differentiation via an ERK1/2-dependent anti-oxidant pathway. GYY4137 protected cells against hydrogen peroxide (H_2_O_2_)-induced cell death and apoptosis.
[83]	Mo et al.	GYY4137	In vivo	Mice	GYY4137 increased the number of osteoclasts in periodontal tissues and tooth movement distance in CSE^−/−^ mice.
[86]	Lee et al.	NaHS	In vitro	Human Mice	NaHS protected hPDLCs from nicotine and LPS-induced cytotoxicity and recovered nicotine- and LPS-downregulated osteoblastic differentiation. NaHS inhibited the differentiation of osteoclasts in mouse bone marrow cells.
[88]	Jiang et al.	NaHS	In vitro	Human	NaHS promoted osteogenic differentiation of human periodontal ligament cells via P38-Mapk Signaling Pathway under proper tension stimulation.
[89]	Pu et al.	NaHS	In vivo	Mice	NaHS increased the orthodontic tooth movement by decreasing bone mineral density and up-regulating osteoclasts.
[90]	Xu et al.	GYY4137	In vivo	Rat	GYY4137 increased bone mineral density in vertebra and prevented bone loss in ovariectomized rats.
[91]	Behera et al.	NaHS	In vivo In vitro	Mice	NaHS prevented hyperhomocysteinemia-induced bone loss, viac-Jun/JNK signaling. NaHS inhibited osteoclasts and induced osteogenesis.
[93]	Yang et al.	NaHS	In vivo	Rats	H_2_S mitigated SCI-induced sublesional bone loss.
[94]	Jiang et al.	GYY4137	In vivo	Rabbits	GYY4137 accelerated osteogenesis during distraction osteogenesis.

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
