# Peer review of "Hydrogen Sulfide in Bone Tissue Regeneration and Repair: State of the Art and New Perspectives"

_ijms, 2019, doi:10.3390/ijms20205231_

Round 1

Reviewer 1 Report

The manuscript by Gambari et al. demonstrates the role of H2S in bone tissue regeneration and repair. Although this manuscript well describes the potential implication of H2S in the above health issue and summarizes current advancement in H2S releasing devices to cure this problem. However, the manuscript needs little more work in explaining the exact biochemical mechanism of how H2S works, perhaps a figure would be better in explaining the following points, such as whether it acts as a scavenger of reactive oxygen species (ROS), if yes then whether any other antioxidants (NAC, vitamin C) works in the same way like H2S does, whether H2S has any role in reducing protein disulfide bonds, if yes then whether any other reducing agents also works same way like H2S does, whether there are any side effects associated with these types of treatments, if yes then how we can overcome them. It will better if they can summarize the findings of different studies about the different H2S donor compounds, their half-life, and which models (in vitro/in vivo) they used.  

Author Response

We thank the reviewer for proving helpful suggestions to improve our review manuscript. In the revised section 1 of the review, we have now added a figure (new Figure 2) which summarizes the main mechanisms by which H2S-donors regulate bone cells and their signaling pathways. The role of H2S as a scavenger of ROS in osteoclasts is now highlighted both in the text (section 1.1) and in Figure 2; moreover, we now mention the study providing the most compelling evidence of a direct link between the antioxidant activity of NAC and mitochondrial H2S generated by NAC treatment. Furthermore, the revised version features a new table (Table 1) which summarizes the main findings of different studies about the different H2S donor compounds, the models in which findings were generated, and the relevant reference to the bibliography.

Reviewer 2 Report

The manuscript title “Hydrogen sulfide in bone tissue regeneration and
3 repair: state of the art and new perspectives", certainly provides a new perspective about the use of H2S in bone tissue regeneration

Below are the comments which authors can include improving the review.

It will more appealing if authors can keep section 2 only about bone regeneration and for example just focus on studies about MSC differentiation and increase in COLI expression rather than talking about a myocardial infraction or skin graft.

The studies can be used to show electrospun membrane have loaded H2S for delivery, but no description needed.

Author Response

We thank the reviewer for the suggestions.

As we point out in the final section of the manuscript, even though the field is growing and gathering increasing attention, applications of H2S-loaded biomaterials to bone regeneration are still scarce. As we approached the section on ‘H2S-releasing scaffolds for regenerative medicine’, we felt it was appropriate to provide a brief description of the biological effects that resulted from the different strategies of incorporation of H2S-donors in various biomaterials. Because these strategies are sometimes rather complex and H2S may be quickly vanishing from biological systems, the evidence of a biological response to a newly designed H2S-releasing material is important to confirm the effectiveness of the chemical strategy and of the manufacturing process. We agree that this section is briefly taking the reader off the main track of the review, but we believe this is justified and provides more robust evidence that those materials could be one day translated into applications for bone regeneration.